# Physician's use of sickness certification guidelines: a nationwide survey of 13 750 physicians in different types of clinics in Sweden

Veronica Svärd ,[1,2] Kristina Alexanderson[1]

[1]Division of Insurance Medicine, Department of Clinical Neuroscience, Karolinska Institutet, Stockholm, Sweden
[2]Department of Social Work, Södertörn University, Huddinge, Sweden

**Correspondence to**
Dr Veronica Svärd;
veronica.svard@ki.se

## ABSTRACT

**Objectives** To explore physicians' experiences of using the national sickness certification guidelines introduced in 2007 and the types of information they used, in general and in different types of clinics.

**Design** Cross-sectional survey.

**Setting** Most physicians working in Sweden in 2017.

**Participants** A questionnaire was sent to 34 718 physicians; 54% responded. Analyses were based on answers from the 13 750 physicians who had sick leave cases.

**Outcome measures** To what extent the guidelines were used and what type of information from them that was used.

**Results** Ten years after the sickness certification guidelines were introduced in Sweden, half of the physicians used them at least once a month. About 40% of physicians in primary healthcare and occupational health services used the guidelines every week. The type of information used varied; 53% used recommendations about duration and 29% about degree of sick leave. Using information about function and activity/work capacity, respectively, was more common within primary healthcare (37% and 38%), psychiatry (42% and 42%), and occupational health services (35% and 41%), and less common in surgery and orthopaedic clinics (12% and 12%) who more often used information about duration (48% and 53%). Moreover, 10% stated that the guidelines were very, and 24% fairly problematic to apply. Half (47%) stated that the guidelines facilitated their contacts with patients and 29% that they improved quality in their management of sick leave cases. More non-specialists, compared with specialists, found that the guidelines facilitated contacts with patients (OR 3.28, 95% CI 3.04 to 3.55).

**Conclusions** The majority of the physicians used the sickness certification guidelines, although this varied with type of clinic. Half stated that the guidelines facilitated patient contacts. Yet, some found it problematic to apply the guidelines. Further development of the guidelines is warranted as well as more knowledge about them among physicians.

## Strengths and limitations of this study

► With 13 750 responding sick-listing physicians, this is the largest survey of physicians' work with sick leave cases, also internationally, allowing several types of subgroup analyses.

► All physicians involved in sickness certification were included, not only general practitioners, which is seldom the case in this research area.

► Although the response rate (54%) was relatively high for this type of study, we do not know if non-responders would have responded differently.

► Comparisons with findings from the few studies from other countries is problematic due to large differences in both sickness absence insurance systems and in available sickness certification guidelines.

► The cross-sectional design means that no conclusions about causality can be drawn.

interventions have been initiated to keep the rates at a more stable and lower level. Several of the interventions have been directed towards physicians, who have a central role in the sick leave process. Systematic reviews have found that physician's find aspects of sickness certification (SC) of patients as problematic,[2–6] for example, the following aspects: their own relative lack of competence regarding insurance medicine, lack of support for handling SC, and the general lack of scientific knowledge regarding optimal sick leave and handling of such work tasks.[2–6] This is one of the reasons for why different types of SC guidelines have been asked for and, in some countries also introduced.[7–11] In Sweden, such guidelines were introduced in 2007.

SC guidelines are to assist clinicians in their handling of consultations regarding sickness absence. These types of guidelines serve, when they exist, as recommendations for clinicians to uphold the best practice regarding SC. Some of the guidelines that have been introduced are, however, merely based on data

## INTRODUCTION

In Sweden, the sick leave rates have fluctuated much in the last decades,[1] and many

regarding duration of previous sick leave cases,[10] which involves several problems when put in practical use. An essential barrier to develop guidelines in Sweden as well as in other countries, is the lack of scientific knowledge regarding the need for, as well as possible consequences/side effects of being on sick leave for different periods of time among patients with specific medical conditions. Moreover, such knowledge would be needed in relation to different types of work task the patient has, as it is not a diagnosis in itself that gives the right to be on sick leave but the extent to which the diagnosis limits functions that are of importance for that individual's work capacity—that is, in relation to his/her specific work tasks.

Due to the limited number of studies regarding optimal and/or needed sick leave among people with different medical conditions and work tasks, such guidelines so far cannot be based on scientific evidence.[8] Although Letrilliart and Barrau in a systematic review about issues regarding SC, argued that one strategy to improve sick-listing practice would be to provide guidelines, they found that general practitioners (GPs) in Europe demonstrate low awareness and use of SC guidelines, and that the guidelines are ineffective without training in how to use them.[9]

Most studies regarding physicians' SC practice have so far focused on GPs.[2 4–6] However, in Sweden, as in many countries, all physicians, not only GPs can issue sick leave certificates. In Sweden, such a certificate is needed after the seventh day of a sick leave spell. The SC is used by the officers of the Social Insurance Agency to assess whether the patient/claimant fulfils the criteria for sick leave benefits, which can be granted for full time or part time (100%, 75%, 50% or 25%) of ordinary work hours. However, as clear from above, there is a nearly total lack of scientific evidence on optimal sick leave duration and degree for patients with specific diagnoses and work tasks. Therefore, the SC guidelines for specific diagnoses were developed in Sweden by the National Board of Health and Welfare, based on discussions in expert groups, rather than based on established scientific evidence. For many diagnoses and for less severe conditions, the recommendation in the guidelines was no sick leave at all, or just for a few days or weeks. For other diagnoses, longer sick leave spells could be accepted, especially if the patient had a physically demanding job. The Swedish National Board of Health and Welfare issued the web-based nationwide SC guidelines in 2007. They consist of two sets of recommendations: general guidelines covering the principles related to SC and the diagnosis-specific recommendations.[12] The latter includes recommendations for duration and degree (full time or part time) of sick leave as well as diagnosis-specific information about assessments of function, activity and work capacity.[13] They cover about 110 of the most prevalent sick leave diagnoses and are continuously updated, based on experiences from physicians and others, inconsistencies or ambiguities in the texts, effects on work capacity from new treatments, continuous expert discussions, etc.

Previous studies about physician's use of these SC guidelines in Sweden show that a large majority (>85%) of GPs who have used guidelines find them beneficial to ensure high quality in handling sick leave cases.[14] The use of guidelines among GPs[14] and among gynaecologists/obstetricians[15] have increased over the years, as did the rate of gynaecologists/obstetricians who reported that the guidelines facilitated their contacts with patients.[15] Another study compared the quality of SC before and after the SC guidelines were introduced, in 2007 respectively 2009.[16] Improvement of the quality of the information provided in the certificates was found regarding descriptions on functioning—although body impairments still dominated the descriptions of functioning, contradictory to the SC guidelines emphasising of that functioning should be described in terms of activity and work incapacity. These previous studies do not, however, provide any knowledge about what type of information from the guidelines physicians find useful, and if this differs between type of clinic, such as psychiatry, orthopaedics, and primary healthcare. Therefore, the aim of this study was to explore physicians' experiences of using the nationwide SC guidelines and what kind of information in the guidelines they used, and how this differed with type of clinic.

## METHODS

Analyses of answers to a questionnaire sent to 34 718 physicians working and living in Sweden.

### The questionnaire

A questionnaire about physician's work with sick leave consultation including 133 questions was developed, based on previous surveys (2004, 2008 and 2012).[17–20] The items in the previous questionnaires were based on results from interviews, pilot studies and other studies.[2 21] Minor revisions of the 2012 questionnaire were made, based on previous results and extensive discussions with physicians and with representatives from physician organisations and other stakeholders.

The questionnaire included questions about frequency of sick leave consultations, frequencies and severity of experienced problems, organisational prerequisites, internal and external cooperation, competence in insurance medicine, and about the use of the Swedish SC guidelines—the latter was the focus of this study.

### Participants

The questionnaire was sent to all the physicians, 68 years and younger, living and working in Sweden in 2017, identified from the Swedish Healthcare Address Register of all physicians working in Sweden, administrated by QuintilesIMS. Specialists working in clinics that seldom have sick leave cases were excluded, for example, in laboratory, radiology, forensic, geriatric and paediatric clinics. Statistics Sweden administrated the data collection, to ensure the respondents anonymous participation, and delivered

an anonymised data file to the project group, including information on sex and age.

The questionnaire and a prepaid response envelope were sent to the participant's home addresses together with a cover letter with information about the project and that participation was voluntary. The questionnaire could be answered either over the web or by paper. Four reminders were sent to those who had not yet responded. The participants provided consent by submitting the filled questionnaire. It took about 30 minutes to fill out.

The study population included 34 718 physicians, of which 34 585 were eligible, and 18 714 of them responded (54.1% response rate). As usual in surveys, the response rate was somewhat higher among women (56.9% among women and 51.6% among men) and older physicians (51.9% among 20–54 years old and 59.2% among 55–67 years old).

## Patient and public involvement
There were no patients involved in this study.

## Statistical analysis
Descriptive statistics were used. Variables about the use of the SC guidelines were analysed and comparisons were made between types of clinics regarding the following items: educational level, type of clinic, frequency of sick leave cases, use of the guidelines, and if the guidelines were experienced as useful. Further variables concerned what information in the guidelines that were used: recommendations about duration and degree, information about function, activity/work capacity or none. Logistic regression was used to calculate crude and adjusted ORs with 95% CIs regarding if the physicians experienced that the SC guidelines facilitated their contacts with patients/make such contacts easier. Adjustments were conducted for being a specialist and for having sick leave cases at least once a week.

In total, 13 750 physicians (78% of the respondents) had sick leave cases at least once a year and those were included in the study. Those not specifying level of education (0.2% of all) or main type of clinic (0.3%, n=46) were not included in analyses related to level of education and type of clinic, respectively. For the specific questions about the SC guidelines, answers were missing for 2.2%–4.9% of respondents and for one question 7.3%. Those were not included in the analyses of the respective question.

## RESULTS
Of the sick-listing physicians, half (50.4%; n=6929) were women and 68% were board-certified specialists. A third (34%) worked in primary healthcare (table 1), where the proportion of board-certified specialists was somewhat lower (61%). The highest proportions of physicians having sick leave cases at least six times a week were in falling order in orthopaedics (74%), occupational health services (72%), rehabilitation (69%), oncology (61%), psychiatry (58%), pain (54%) and primary healthcare (45%).

Half (50%) used the SC guidelines at least once a month and 26% at least once a week. The frequency in use varied with type of clinic; physicians in occupational health services used the guidelines most often; 18% at least six times a week and 42% at least once a week. This was expected as all their patients are of working age and employed. About the same proportion of the GPs (41%) used the guidelines at least once a week.

Those who had sick leave cases often also used the guidelines more often (figure 1). The proportion (22%–23%) who did not use the guidelines were the same irrespective if the frequency of sick leave cases were 1–5 times a week or more than six times a week. Among those who had such consultations less than once a week, 40% stated that they did not use the guidelines.

Regarding what type of information from the guidelines the physicians used, more than half (53%) used recommendations about duration of sick leave, while 29% used recommendations about degree (part-time or full-time sick leave) (table 1). Half or more of the physicians within primary healthcare (69%), occupational health services (59%), internal medicine (53%), orthopaedics (53%) and infection (50%) used the recommendations about duration. Using recommendations about degree was most common in primary healthcare (43%), gynaecology (33%) and occupational health services (36%). Use of the information about limitation of function or about activity limitation/work capacity was more common within primary healthcare (37% and 38%, respectively), psychiatry (42% and 42%) and occupational health services (35% and 41%). A majority of the physicians within eye (67%), skin (59%), oncology (50%), pain (54%), and ear, nose and throat clinics (52%) did not use any information from the guidelines, partly due to lack of SC guidelines regarding several diagnoses handled in those clinics. Also, the proportions varied within types of clinics regarding type of information used. While about half of the physicians within surgery and orthopaedics used information about duration, only 12% used information about limitation of function or activity/work capacity (table 1). Within psychiatry, on the other hand, almost the same proportions used information about limitation of function or activity/work capacity (42%) as recommendations about the duration (45%).

Ten per cent of those who used the guidelines stated that the they were very problematic to apply and 24% that they were fairly problematic to apply, while 25% never found them problematic to apply. A somewhat higher proportion of those who used the guidelines less than once a week (11%) compared with at least once a week (8%) stated that they were very problematic to use (figure 2).

Nearly one-third (29%) reported that the guidelines to a high extent improved the quality in their management of SC; and higher proportions in gynaecology (35%) and internal medicine (33%) stated this, compared with, for

**Table 1** Number and proportion (%) of sick-listing physicians, proportions (%) of frequency of sick leave cases, of frequency in use and of types of information in the sickness certification guidelines used, by type of clinic (falling order according to size)

| Type of clinic | n | %* | Frequency of sick leave cases | | | Frequency in using the guidelines | | | | | Type of information used from the guidelines | | | | | |
| --- | --- | --- | --- | --- | --- | --- | --- | --- | --- | --- | --- | --- | --- | --- | --- | --- |
| | | | ≥6 cases a week | 1–5 cases a week | Monthly | At least six times a week | 1–5 times a week | Monthly | A few times a year | Never/almost never | None | Duration | Part-/full-time | Function | Activity/work capacity | Other |
| Primary healthcare | 4687 | 34.1 | 45 | 50 | 4 | 9 | 32 | 31 | 18 | 10 | 18 | 69 | 43 | 37 | 38 | 3 |
| Internal medicine | 1661 | 12.1 | 12 | 46 | 41 | 2 | 10 | 22 | 34 | 32 | 37 | 53 | 28 | 19 | 18 | 2 |
| Surgery | 1285 | 9.3 | 27 | 52 | 21 | 4 | 13 | 18 | 27 | 38 | 46 | 48 | 18 | 12 | 12 | 1 |
| Psychiatry | 1050 | 7.6 | 58 | 32 | 10 | 11 | 23 | 27 | 20 | 19 | 28 | 45 | 29 | 42 | 42 | 3 |
| Other | 1029 | 7.5 | 33 | 20 | 36 | 4 | 11 | 18 | 24 | 43 | 44 | 37 | 20 | 16 | 17 | 1 |
| Orthopaedic | 921 | 6.7 | 74 | 23 | 3 | 11 | 21 | 20 | 21 | 27 | 39 | 53 | 15 | 12 | 12 | 2 |
| Ear, nose and throat | 421 | 3.1 | 9 | 56 | 35 | 0 | 3 | 13 | 30 | 53 | 52 | 38 | 17 | 14 | 14 | 2 |
| Infection | 357 | 2.6 | 9 | 55 | 36 | 1 | 7 | 18 | 34 | 41 | 44 | 50 | 20 | 10 | 10 | 0 |
| Oncology | 342 | 2.5 | 61 | 31 | 8 | 5 | 11 | 20 | 25 | 39 | 50 | 34 | 13 | 16 | 15 | 2 |
| Eye | 341 | 2.5 | 3 | 19 | 78 | 1 | 3 | 8 | 28 | 60 | 67 | 25 | 9 | 8 | 9 | 1 |
| Occupational health services | 338 | 2.5 | 72 | 20 | 8 | 18 | 24 | 22 | 22 | 13 | 22 | 59 | 36 | 35 | 41 | 4 |
| Gynaecology | 320 | 2.3 | 21 | 51 | 28 | 5 | 19 | 22 | 26 | 27 | 37 | 45 | 33 | 18 | 21 | 1 |
| Neurology | 305 | 2.2 | 29 | 58 | 13 | 1 | 12 | 16 | 35 | 36 | 46 | 38 | 23 | 20 | 19 | 2 |
| Rheumatology | 198 | 1.4 | 24 | 61 | 16 | 2 | 13 | 17 | 31 | 37 | 46 | 36 | 23 | 23 | 24 | 3 |
| Skin | 171 | 1.2 | 4 | 8 | 88 | 1 | 2 | 6 | 37 | 54 | 59 | 35 | 15 | 11 | 12 | 1 |
| Rehabilitation | 158 | 1.1 | 69 | 23 | 7 | 10 | 14 | 15 | 35 | 25 | 40 | 40 | 18 | 23 | 25 | 3 |
| Administration† | 66 | 0.5 | 9 | 30 | 60 | 3 | 10 | 13 | 25 | 49 | 45 | 39 | 24 | 18 | 26 | 2 |
| Pain | 54 | <0.5 | 54 | 33 | 13 | 10 | 4 | 14 | 24 | 47 | 54 | 22 | 17 | 24 | 19 | 4 |
| All physicians | 13 750 | 100 | 38 | 43 | 19 | 6 | 20 | 24 | 24 | 26 | 33 | 53 | 29 | 25 | 26 | 2 |

*Column percentage, all other row percentages.
†Including research and education (ie, physicians mainly working with education, research or as managers but also having patients some of the time).

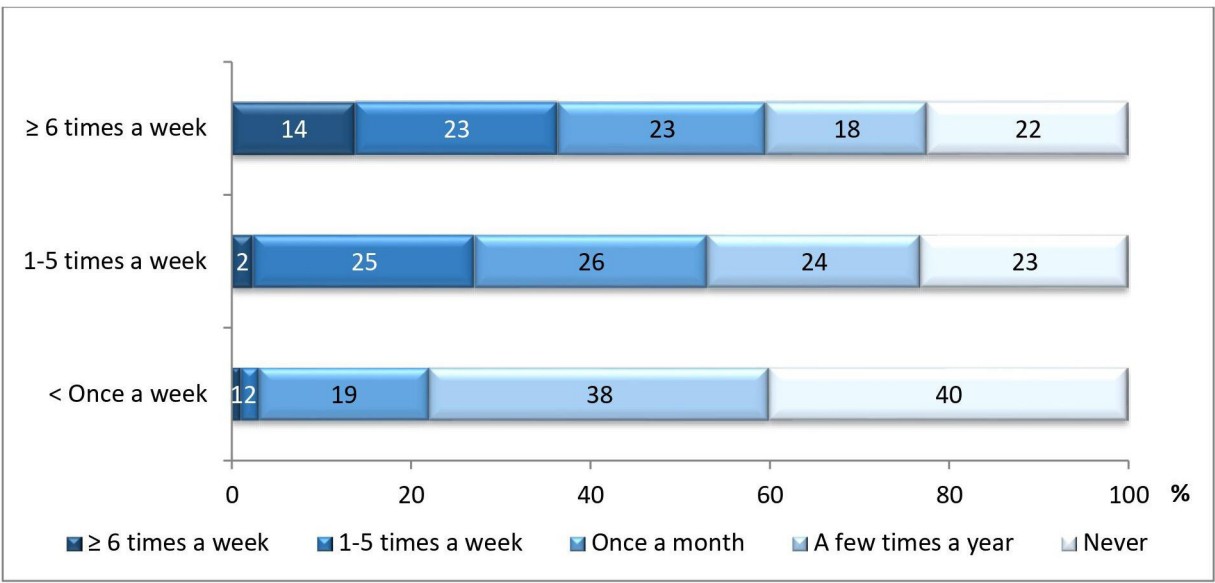

**Figure 1** Proportion (%) of physicians who used the sickness certification guidelines with different frequencies (x-axis), by how often they had sick leave cases (y-axis).

example, in the clinics with fewer physicians such as rehabilitation (17%) and pain clinics (19%) (table 2).

Half (53%) stated that the guidelines facilitated their contacts, or made the contacts easier, with at least one of the suggestions, that is, patient, Social Insurance Agency, healthcare staff, patients' employer and/or employment office. About twice as high proportions in primary healthcare (65%) and occupational health services (58%) stated this compared with among the physicians in smaller clinics that seldom used the guidelines, such as eye clinics (27%). Lower proportions stated that the guidelines facilitated their contacts with employers or employment offices (21%) and healthcare staff (29%) compared with with the Social Insurance Agency (39%) and patients (47%). However, this varied with type of clinic. Here it is important to remember that all physicians had contacts with their patients, while only a few had contacts with the other stakeholders, which is why the figures for those are expected to be lower. A third

(31%) of physicians in occupational health services stated that the guidelines facilitated their contacts with patients' workplaces. Among those mainly working in oncology (37%), psychiatry (43%) or neurology (32%), the highest proportions stated that the guidelines facilitated their contacts with the Social Insurance Agency. For most type of clinics, however, the highest proportions reported that the guidelines facilitated their contacts with the patient.

The ORs for stating that the SC guidelines facilitated the contact with the patient was higher among women (OR 1.23 (95% CI 1.15 to 1.32) then men, among younger physicians (OR aged 24–39: 3.13 (2–86–3.42); OR aged 40–54: 1.38 (1.26–1.51)), compared with the older physicians (aged 55–68). Also, the OR regarding this was more than three times as high for non-specialists (OR 3.28 (3.04–3.55)), compared with specialists. As shown in table 2, a higher rate of the physicians in primary healthcare stated that the guidelines facilitated contacts with patients (55%). In table 3, the ORs for stating this

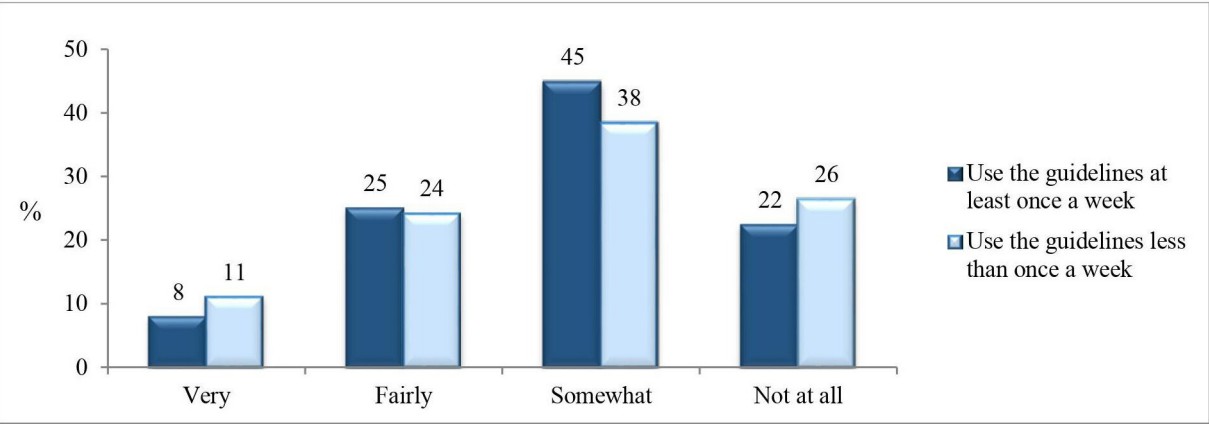

**Figure 2** Proportions (%) of physicians who stated it was problematic to apply the sickness certification guidelines, by how often they used the guidelines.

**Table 2** Proportion (%) of physicians who stated that the sickness certification guidelines are important for good quality in her/his handling of sickness certification and facilitate their contacts with patients or others, by type of clinic

| Type of clinic | The extent to which guidelines is of importance for the quality of how I handle sickness certification | | | The guidelines facilitate my contacts with… | | | | | |
| | Very | Moderately | None | The patient | Social Insurance Agency | Healthcare staff | Employer | Employment Office | At least one of all |
|---|---|---|---|---|---|---|---|---|---|
| Primary healthcare | 31 | 57 | 13 | 55 | 45 | 33 | 23 | 25 | 65 |
| Internal medicine | 33 | 51 | 16 | 41 | 41 | 31 | 22 | 22 | 53 |
| Surgery | 30 | 49 | 21 | 46 | 34 | 28 | 18 | 17 | 48 |
| Psychiatry | 30 | 54 | 16 | 40 | 43 | 31 | 22 | 25 | 55 |
| Other | 30 | 45 | 25 | 35 | 35 | 30 | 19 | 20 | 39 |
| Orthopaedic | 24 | 51 | 25 | 44 | 35 | 24 | 16 | 16 | 50 |
| Ear, nose and throat | 23 | 56 | 22 | 39 | 31 | 22 | 17 | 18 | 41 |
| Infection | 32 | 54 | 15 | 44 | 35 | 23 | 18 | 16 | 46 |
| Oncology | 24 | 55 | 21 | 35 | 37 | 20 | 13 | 13 | 42 |
| Eye | 21 | 56 | 23 | 26 | 25 | 25 | 13 | 14 | 27 |
| Occupational health services | 22 | 62 | 16 | 46 | 37 | 24 | 31 | 17 | 58 |
| Gynaecology | 35 | 51 | 15 | 48 | 37 | 29 | 19 | 16 | 51 |
| Neurology | 21 | 59 | 20 | 31 | 32 | 17 | 12 | 14 | 42 |
| Rheumatology | 21 | 49 | 30 | 34 | 33 | 24 | 19 | 19 | 41 |
| Skin | 26 | 54 | 20 | 40 | 27 | 24 | 15 | 16 | 40 |
| Rehabilitation | 17 | 63 | 20 | 32 | 30 | 22 | 19 | 19 | 42 |
| Administration | 21 | 55 | 23 | 33 | 36 | 18 | 18 | 16 | 45 |
| Pain | 19 | 57 | 31 | 33 | 28 | 28 | 18 | 20 | 33 |
| All physicians | 29 | 54 | 17 | 47 | 39 | 29 | 21 | 21 | 53 |

is shown for five specific clinical settings, using the 4477 Other physicians as reference group. For all types of clinics, ORs are statistically significant, with the exception of psychiatry, also when adjusting for being a specialist and for having sick leave cases at least once a week. The GP's OR was especially high (adjusted OR 1.78 (95% CI 1.62 to 1.95). When restricting those analyses to the physicians who at least once a month used the guidelines, the results differed; after adjustments the OR in the group Other physicians (n=1415) was somewhat higher than in several of the other clinics, however, only significantly so for those in psychiatric clinics.

## DISCUSSION

In this study, a very large number (13 750) of the sick-listing physicians in Sweden responded to questions about their use of the SC guidelines, ten years after the guidelines were introduced in Sweden in 2007. Half (50%) used the guidelines at least once a month and 26% at least once a week, although those rates differed much with type of clinic and also regarding what type of

information the physician used. Half (47%) stated that the guidelines facilitated their contacts with patients regarding SC and discussion about sick leave. Furthermore, 29% of the physicians stated that the guidelines improved quality in their management of SC cases. Ten per cent stated that the guidelines were very problematic to apply and 17% did not experienced that the guidelines improved the quality of their handling of SC.

Comparing results with those from previous Swedish studies shows that the every-week use increased among GPs from 22% in 2008[11] and 29% in 2012[14] to 41% in 2017, that is, in this survey. Among occupational health physicians, every week use increased from 33% in 2008[22] to 42% in 2017. Internationally, there are hardly no equivalent studies, however, in the UK it was found that 4 years after introduction of SC guidelines, 36% of the 77 responding GPs were aware of the guidelines and 20% of these 77 GPs used them.[23] This might be compared with the 76% of the responding GPs who used guidelines at least a few times a year already 1 year after the guidelines were introduced in Sweden in 2007.[11] While Letrilliart

**Table 3** Crude and adjusted OR with 95% CIs for stating that the sickness certification guidelines facilitate the communication with patients, in different type of clinics

| Type of clinic | Crude OR | 95% CI | Adjusted OR* | 95% CI | Adjusted OR† | 95% CI | Adjusted OR‡ | 95% CI |
|---|---|---|---|---|---|---|---|---|
| **All** | | | | | | | | |
| All other physicians (n=4477) | Ref | | Ref | | Ref | | Ref | |
| Internal medicine | 1.51 | 1.35 to 1.70 | 1.46 | 1.30 to 1.65 | 1.49 | 1.32 to 1.67 | 1.43 | 1.27 to 1.62 |
| Psychiatry | 1.02 | 0.89 to 1.97 | 1.03 | 0.89 to 1.19 | 1.05 | 0.91 to 1.21 | 1.07 | 0.92 to 1.24 |
| Orthopaedic | 1.17 | 1.01 to 1.17 | 1.25 | 1.07 to 1.45 | 1.22 | 1.05 to 1.41 | 1.31 | 1.12 to 1.53 |
| Occupational health services | 1.30 | 1.04 to 1.35 | 1.82 | 1.45 to 2.28 | 1.22 | 1.05 to 1.41 | 1.90 | 1.51 to 2.39 |
| Primary healthcare | 1.81 | 1.66 to 1.63 | 1.70 | 1.56 to 1.85 | 1.88 | 1.72 to 2.05 | 1.78 | 1.62 to 1.95 |
| **Used the guidelines at least monthly** | | | | | | | | |
| All other physicians (n=1415) | Ref | | Ref | | Ref | | Ref | |
| Internal medicine | 1.29 | 1.04 to 1.60 | 1.23 | 0.98 to 1.53 | 1.17 | 0.94 to 1.46 | 1.12 | 0.90 to 1.41 |
| Psychiatry | 0.53 | 0.44 to 0.65 | 0.56 | 0.46 to 0.68 | 0.55 | 0.46 to 0.67 | 0.58 | 0.48 to 0.71 |
| Orthopaedic | 0.93 | 0.74 to 1.16 | 1.00 | 0.80 to 1.26 | 0.99 | 0.79 to 1.24 | 1.07 | 0.85 to 1.34 |
| Occupational health services | 0.67 | 0.50 to 0.89 | 0.90 | 0.67 to 1.21 | 0.71 | 0.53 to 0.95 | 0.96 | 0.71 to 1.29 |
| Primary healthcare | 0.93 | 0.81 to 1.05 | 0.94 | 0.82 to 1.07 | 0.98 | 0.86 to 1.12 | 0.99 | 0.86 to 1.13 |

Among all physicians and among physicians who at least once a month used the guidelines.
*Adjusted for specialists.
†Adjusted for having sick leave cases at least once a week.
‡Adjusted for both* and †.

and Barrau systematic review found that GPs in Europe demonstrate low awareness and use of SC guidelines,[9] our findings show that this is not the case in Sweden. However, precise comparisons cannot be made between the studies, conducted in different European countries, because of different selections, contexts, and regulations; yet they show that differences between the countries exists. One reason for the high awareness and use of the SC guidelines in Sweden might be that the Social Insurance Agency use the guidelines when assessing claimant's right to sick leave benefits. The relatively high use among physicians in occupational health services should be related to that they only have employed patients, while other physician to a larger extent have patients who not are in paid work, or not even of working age.

For some groups of diagnoses there are fewer diagnosis-specific SC guidelines than for other groups, which probably explains some of the large variations between different type of clinics' use of the guidelines. Another finding was that the type of information used varied somewhat with type of clinic. In psychiatry, a comparatively high proportion used information about function and activity/work capacity, perhaps meaning these guidelines were particularly well written, or that psychiatrists need such information more than other physicians. In other type of clinics, such as internal medicine, neurology, gynaecology and pain, information about function limitations were used to a higher extent than information about activity/work capacity, perhaps indicating a higher need of such information there.

While some previous studies state that SC guidelines have not improved clinical practice,[9 24] our findings suggest otherwise regarding the Swedish SC guidelines. Slightly more than half (55%) of the GPs in our study found that the guidelines facilitated their contacts with patients in sick leave cases. Considering that our study included the responses from all participating GPs, and not only those 90% who answered that they had used the SC guidelines as in the study by Gustavsson et al,[14] the proportion who stated that the guidelines facilitate their contacts with patients seem to have been fairly stable over time. In 2012, 61% of those who had used the SC guidelines reported that the guidelines facilitated their contacts with patients, and 56% reported this in 2008,[14] when the guidelines were newly implemented. Moreover, in analyses of ORs for stating that the guidelines facilitated their contacts with patients, among all sick-listing physicians, showed higher ORs among women (OR 1.23 (95% CI 1.15 to 1.32)), younger physicians (OR aged 24–39: 3.13 (2–86–3.42); OR aged 40–54: 1.38 (1.26–1.51)), as well as for non-specialists (OR 3.28 (95% CI 3.04 to 3.55)). This is in line with results from Skånér et al[11] that non-specialists within primary healthcare to a larger extent stated that the guidelines facilitated their contacts with patients. Moreover, the adjusted ORs for experiencing that the SC guidelines facilitated the physician's contacts with patients was highest among GPs. Comparing results with regards to if SC guidelines are perceived to be of importance for the quality of the management of SC, 13% of the GPs in our study found the guidelines not

to be important at all, a higher proportion than the 9% Skånér *et al* found in 2012.[11]

Strength of the study are that nearly all physicians, not only a sample or a specific type of physicians (ie, GPs), were invited, and the large number of respondents, allowing several subgroup analyses, for example, types of clinics or level of education. Although the response rate (54%) was relatively high, a limitation is that we do not know if non-responders vary regarding the here studied questions. Nevertheless, that is improbable as the questions regarding guidelines came late in the comprehensive questionnaire of 133 questions. It is possible that physicians who experience themselves being too busy to answer the questionnaire, also experience themselves being too busy to search for and use the SC guidelines. However, we might also have an underestimation in rates of physicians stating they used the guidelines. As the guidelines had existed for 10 years when answering the survey, some physicians who answer that they do not use guidelines, may state so because they are already familiar with the guidelines and the recommendations, and thus no longer actively check the web-site regarding, for example, sick leave duration or degree for a specific diagnosis. The fact that the guidelines has become a part of many physicians daily SC practice also means they may value them less highly than when they were introduced ten years before, due to that the guidelines are not perceived to add any new information, in the way they previously might have done. The results of this study could be generalised to Sweden and probably other countries that have introduced these types of SC guidelines, especially those with similar sickness absence and SC systems.

### Implications for practice
Knowledge about use, usefulness and problems regarding aspects of the SC guidelines is warranted to be able to improve their usefulness for different types of clinics. The finding that there were several types of clinics where a majority of physicians handled sick leave cases every week but seldom used the guidelines, indicates that there might be a need for developing the guidelines, for example, with regard to more diagnoses, or more useful and updated information. A clinical implication elucidated is that less experienced physicians, especially in clinical settings such as primary healthcare where sick leave cases are common, should be provided information about the SC guidelines in workplace introductions.

### Conclusions and suggestions for further research
The lack of scientific knowledge concerning the use and relevance of SC guidelines is remarkable, given the considerable economic costs to patients, employers, healthcare and society due to sick leave. Physicians in different types of clinics use different types of information in the guidelines, indicating varied needs. Further research should focus on factors that hinder and promote the use of SC guidelines and on what information physicians need from such guidelines, in general as well as in different types of clinics.

**Correction notice** This article has been corrected since it was published. The footnote symbols in Table 1 were incorrectly linked. This has now been corrected.

**Contributors** KA initiated the project including the data collection, made the study design, developed the questionnaire, and is the guarantor. A statistician performed the data management and statistical analyses. VS drafted the manuscript and the tables. Both authors participated in presentation and interpretation of data and reviewed the manuscript.

**Funding** The data collection was funded by the Swedish Social Insurance Agency (066754-2016). The Agency was not involved in the design of this study, analysis, interpretation of the data, nor as authors of the manuscript. The authors have had access to all data.

**Competing interests** None declared.

**Patient and public involvement** Patients and/or the public were not involved in the design, or conduct, or reporting, or dissemination plans of this research.

**Patient consent for publication** Not applicable.

**Ethics approval** This study involves human participants and was approved by The Regional Ethical Review Board of Stockholm, Sweden (2017/95-32).

**Provenance and peer review** Not commissioned; externally peer reviewed.

**Data availability statement** Data may be obtained from a third party and are not publicly available. The data used in this study are administered by the Division of Insurance Medicine, Karolinska Institutet, and cannot be made publically. According to the General Data Protection Regulation, the Swedish law SFS 2018:218, the Swedish Data Protection Act, the Swedish Ethical Review Act, and the Public Access to Information and Secrecy Act, data can only be made available, after legal review, for researchers who meet the criteria for access to this type of confidential data. Readers may contact KA (kristina.alexanderson@ki.se).

**ORCID iD**
Veronica Svärd http://orcid.org/0000-0002-3868-0254

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
