## [Reviewer comments · BMJ Open]

ARTICLE DETAILS

TITLE (PROVISIONAL)	Physician's use of sickness certification guidelines: a nationwide survey of 13 750 physicians in different types of clinics in Sweden
AUTHORS	Svärd, Veronica; Alexanderson, Kristina

VERSION 1 – REVIEW

REVIEWER	Nantha, Yogarabindranath Monash University Malaysia
REVIEW RETURNED	07-Jun-2021

GENERAL COMMENTS	This study showcases trends seen in the uptake and use of sickness certification (SC) guidelines in Sweden. The representativeness of the survey is enhanced by the inclusiveness (involving participants from almost every strand of the medical discipline) and breadth (participants from the whole of Sweden) of the study. Descriptive for the most part, the author makes partial use of logistic regression analysis to demonstrate that [sickness certification guidelines facilitated the contact with the patient] (the statement in parenthesis appears highly confusing in the main text, lacking clarity as to what it exactly means). The manuscript was well written and undoubtedly comprehensive. However, my main point of contention is simply this – at best, this research can be seen as just an audit of patterns seen amongst clinicians in relation to the awareness (and subsequent utilization) of SC guidelines. The study leaves much to be desired in this respect. Moreover, these trends are widely available in other countries (the UK, for example), as alluded to by the author in the main text. In other countries (where data “appear” to be absent or lacking), these findings do not see the light of day primarily because 1) they form a rudimentary quality assurance assessment at healthcare institutions, 2) they are stored as volumes of routine audit documents in healthcare system archives, and therefore 3) remain unpublished (but available upon request). That said, all is not lost. The authors turn to inferential statistics towards the end of the results section, neatly documenting specific odds ratios from a logistic regression analysis. Here, the research details become strikingly interesting. But alas, this intrigue was fairly short-lived, shadowed by the doubt created, in due part, to a hurried and highly simplistic reporting of logistic regression analysis. The reader does not catch a glimpse of the explanatory power of the model (Nagelkerke R square value), nor was there any explicit mention of the controlled variables for this analysis (apart from “the number of specialists”). It can be argued that many other important variables should be controlled for (years of
---

	experience, for example), yielding a more accurate representation of the data used here. Therefore, the description of the logistic regression analysis requires more elaborate and precise details. Here is one good example: Direct logistic regression was performed to assess the impact of a number of factors on the likelihood that respondents would report that they had a problem with their sleep. The model contained five independent variables (sex, age, problems getting to sleep, problems staying asleep and hours of sleep per weeknight). The full model containing all predictors was statistically significant, $X^2(5, N=241) = 76.02, P < .001$, indicating that the model was able to distinguish..... There are increasing calls in research asking deep and meaningful questions. To that effect, I felt the authors could have asked insightful questions and use that information to convince the reader/policymakers of the importance of these findings. These questions should have gone beyond above and beyond improving the “usefulness” of the SC guidelines. Instead, we should ask ourselves the following: 1) if there is increased use of SC guidelines in Sweden, how does that translate to any impact on overall patient care?; 2) do clinicians only use SC guidelines (as mentioned by the author) only when there is a compelling reason to do so (made compulsory by insurance companies)?, or; 3) are clinicians intrinsically motivated to use SC guidelines without being nudged to do so? I miss what novelty this paper could bring to what is already known about sickness certification in the field of research other than just figures about trends in SC guideline utilization. With that, I hope the authors will find my comments useful to increase the quality of the material written in this manuscript.
--	---

REVIEWER	Gabbay, Mark University of Liverpool, Health Services Research
REVIEW RETURNED	27-Sep-2021

GENERAL COMMENTS	Lines 43-5 not sure what you mean by 'that fell well out' is this a translation issue or a typo/editing one? Line 59 continuously updated- based on what, evidence, consensus meetings, Feedback loops on RTW outcomes? How are these updates organised and what are the governance arrangements Methods- any estimate of how long in average is required to complete the survey- there are a lot of items? Is it possible those too busy to do so also less likely to read or refer to absence guidance? Please give figures for response rates rather than 'somewhat higher' % of women vs men, proportion in early mid late career or whatever. The descriptive statistics are relevant and clear with tables and narrative. Albeit with the pandemic, a descriptive paper on this survey undertaken 4 years ago seems a bit long, presumably rejected elsewhere- if so have the authors addressed feedback from previous submissions? What was it and how have they revised the paper?
--

	Discussion covers key elements. With this level of response rate, even if non of the non-responders used the guidelines or found them useful it's still a healthy impact!
--	---

VERSION 1 – AUTHOR RESPONSE

Reviewer: 1

Dr. Yogarabindranath Nantha, Monash University Malaysia

Comments to the Author:

Reviewer comment: This study showcases trends seen in the uptake and use of sickness certification (SC) guidelines in Sweden. The representativeness of the survey is enhanced by the inclusiveness (involving participants from almost every strand of the medical discipline) and breadth (participants from the whole of Sweden) of the study.

Authors' response: Thank you for these comments about the comprehensiveness of the study.

Reviewer comment: Descriptive for the most part, the author makes partial use of logistic regression analysis to demonstrate that [sickness certification guidelines facilitated the contact with the patient] (the statement in parenthesis appears highly confusing in the main text, lacking clarity as to what it exactly means).

Authors' response: Thank you for alerting us on that the question regarding experiencing that the guidelines facilitated the contact with the patient was unclear. In Swedish, the term is 'underlätta' which we usually translate by 'facilitate'. Other words that could have been used here are 'The guidelines make the contacts/discussions with the patient regarding sickness certification matters easier'. The same goes for the same question regarding contacts with other stakeholders. Already from the first pilot studies of introducing the guidelines, a high rate of physicians stated in interviews and questionnaires that the guidelines made discussions about sickness certification with patients (and others) easier. We have now made amendments throughout the manuscript to clarify this.

Reviewer comment: The manuscript was well written and undoubtedly comprehensive. However, my main point of contention is simply this – at best, this research can be seen as just an audit of patterns seen amongst clinicians in relation to the awareness (and subsequent utilization) of SC guidelines. The study leaves much to be desired in this respect. Moreover, these trends are widely available in other countries (the UK, for example), as alluded to by the author in the main text. In other countries (where data “appear” to be absent or lacking), these findings do not see the light of day primarily because 1) they form a rudimentary quality assurance assessment at healthcare institutions, 2) they are stored as volumes of routine audit documents in healthcare system archives, and therefore 3) remain unpublished (but available upon request).

Authors' response: Yes, as you state, this is an explorative study, based on a large number of responses to a survey administered to all physicians of relevance in a whole country, not a sample. We do believe that a comprehensive nationwide intervention, such as introducing and implementing SC guidelines in a whole country, is an important topic to gain scientific knowledge about, also from the perspective of physicians. The perspective of others, e.g., patients, need to be explored in other studies.

You might be right that the use of SC guidelines is part of rudimentary quality assurance assessments in many healthcare systems on various countries. However, that is not the case in Sweden, nor in

other countries we collaborate scientifically with about this. As researchers we, moreover, see it as important that results are scientifically published and thus available for others. Our scientific reports published in Swedish have been and are widely used by healthcare organizations in the whole country.

Reviewer comment: That said, all is not lost. The authors turn to inferential statistics towards the end of the results section, neatly documenting specific odds ratios from a logistic regression analysis. Here, the research details become strikingly interesting. But alas, this intrigue was fairly short-lived, shadowed by the doubt created, in due part, to a hurried and highly simplistic reporting of logistic regression analysis. The reader does not catch a glimpse of the explanatory power of the model (Nagelkerke R square value), nor was there any explicit mention of the controlled variables for this analysis (apart from “the number of specialists”). It can be argued that many other important variables should be controlled for (years of experience, for example), yielding a more accurate representation of the data used here. Therefore, the description of the logistic regression analysis requires more elaborate and precise details. Here is one good example:

Direct logistic regression was performed to assess the impact of a number of factors on the likelihood that respondents would report that they had a problem with their sleep. The model contained five independent variables (sex, age, problems getting to sleep, problems staying asleep and hours of sleep per weeknight). The full model containing all predictors was statistically significant, $X^2(5, N=241) = 76.02, P < .001$, indicating that the model was able to distinguish....

There are increasing calls in research asking deep and meaningful questions. To that effect, I felt the authors could have asked insightful questions and use that information to convince the reader/policymakers of the importance of these findings. These questions should have gone beyond above and beyond improving the “usefulness” of the SC guidelines. Instead, we should ask ourselves the following: 1) if there is increased use of SC guidelines in Sweden, how does that translate to any impact on overall patient care?; 2) do clinicians only use SC guidelines (as mentioned by the author) only when there is a compelling reason to do so (made compulsory by insurance companies)?, or; 3) are clinicians intrinsically motivated to use SC guidelines without being nudged to do so? I miss what novelty this paper could bring to what is already known about sickness certification in the field of research other than just figures about trends in SC guideline utilization.

With that, I hope the authors will find my comments useful to increase the quality of the material written in this manuscript.

Authors' response: Thank you for this lively description of your reading, as well as the suggestion of the three other research questions, questions that, however, require other study designs and data. We indeed hope that our results inspire for such studies to be conducted.

You are raising questions about causality in cross-sectional studies as this, something that always is tricky. We rather find it important, for the sake of physicians and decision makers in Sweden and other countries, to explore the use and usefulness of these type of SC guidelines as experienced by physicians themselves. We did not adjust for both age and being a specialist in respective type of clinical setting nor for, e.g., mean number of years having worked there as that probably would introduce over-adjustments. Board certified specialists tend to be older than those not yet specialists and also to have more years of professional experience. It is always important, as you state, to carefully consider what variables to adjust for and how to categorize the adjustment factors. Nevertheless, in line with your suggestions, we conducted additional analyses, presented in the new Table 3. We have now adjusted for specialist as well as for having sick-leave cases at least once a week, and for both. This gives more depth to the analyses.

Reviewer: 2

Prof. Mark Gabbay, University of Liverpool

Comments to the Author:

Reviewer comment: Lines 43-5 not sure what you mean by 'that fell well out' is this a translation issue or a typo/editing one?

Authors' response: Thank you for noticing this typo, which is now deleted.

Reviewer comment: Line 59 continuously updated- based on what, evidence, consensus meetings, Feedback loops on RTW outcomes? How are these updates organised and what are the governance arrangements.

Authors' response: The SC guidelines are continuously updated by the National Board of Health and Welfare; initially based on reactions from physicians on what was experienced as unrealistic (e.g., regarding the initial guidelines for Hepatitis C), and later regarding inconsistencies in the texts, regarding effects of new treatments, e.g., effects of treatments on work capacity, etcetera. As the scientific knowledge regarding expected level and duration of work incapacity in different types of jobs still is very limited, most of the recommendations so far are based on joint expert discussions. However, the National Board of Health and Welfare has in the last years directed more resources to the development of the SC guidelines. More text about this is now included in the manuscript's Introduction section.

Reviewer comment: Methods- any estimate of how long in average is required to complete the survey- there are a lot of items? Is it possible those too busy to do so also less likely to read or refer to absence guidance?

Authors' response: The questionnaire includes 133 items, covering eleven different topics. Physicians state that it took them about 30 minutes to complete the questionnaire. For some, not all questions were relevant, thus taking less time. One of the topics covered different aspects of not having enough time for SC-related work tasks. Unfortunately, we do not have the possibility to explore your assumption that those not responding might to a less extent have taken time to learn about the SC guidelines. This might of course be the case – or the other way around. To learn about what the SC guidelines recommend for the diagnoses you mainly work with goes very fast, a few clicks on the internet. Moreover, often your patient knows about it. Also, this information is sometimes required when you fill out the sickness certificate/sick note.

So far we do not know how the drop outs would have responded even if we and others can have many ideas and hypothesis about the drop outs both in relation to the questions regarding SC guidelines as well as to the other types of items. The list of them could be very long, and still miss important aspects, why we abstain from such speculations in the manuscript. The main thing is that we have a sufficient higher number of participating physicians to provide results that can be stated to be of relevance.

Reviewer comment: Please give figures for response rates rather than 'somewhat higher' % of women vs men, proportion in early mid late career or whatever.

Authors' response: We agree, and have now added the precise figures:

“As usual in surveys, the response rate was somewhat higher among women (56.9% among women and 51.6% among men) and older physicians (51.9% among 20-54 years old and 59.2% among 55-67 years old).”

Reviewer comment: The descriptive statistics are relevant and clear with tables and narrative.

Authors' response: Thank you!

Reviewer comment: Albeit with the pandemic, a descriptive paper on this survey undertaken 4 years ago seems a bit long, presumably rejected elsewhere- if so have the authors addressed feedback from previous submissions? What was it and how have they revised the paper?

Authors' response: We wish we would have had the resource to conduct this study at once when the data were available and cleaned! However, we prioritized to publish and present results in Swedish, which also is what we had funding for. So far, we have published eight different and very comprehensive Swedish reports on the data, each covering between 100 – 260 pages, on the request of different authorities or organizations. However, this the first time we have submitted this manuscript, it has not previously been submitted to, nor rejected by, any other journal. Only one other study based on the survey has been conducted, covering another aspect (GP's contacts with employers), just recently online ahead of print.

(Nordling P, Alexanderson K, Hensing G, Lytsy P. Factors associated with general practitioners' contacts with sick-listed patients' employers - a Swedish nationwide questionnaire study. Scandinavian Journal of Public Health. 2021 (1-9) <https://doi.org/10.1177/14034948211053141> Online ahead of print.)

Reviewer comment: Discussion covers key elements. With this level of response rate, even if non of the non-responders used the guidelines or found them useful it's still a healthy impact!

Authors' response: Thank you for this comment, we agree, as stated above.

VERSION 2 – REVIEW

REVIEWER	Gabbay, Mark University of Liverpool, Health Services Research
REVIEW RETURNED	22-Nov-2021

GENERAL COMMENTS	Many thanks, I am happy with the revised submission
---